# Early prediction of disease progression in COVID-19 pneumonia patients with chest CT and clinical characteristics

Zhichao Feng [1], Qizhi Yu[2,3], Shanhu Yao[1], Lei Luo[1], Wenming Zhou[4], Xiaowen Mao[5], Jennifer Li [6], Junhong Duan[1], Zhimin Yan[1], Min Yang[1], Hongpei Tan[1], Mengtian Ma[1], Ting Li[1], Dali Yi [1], Ze Mi[1], Huafei Zhao[1], Yi Jiang[1], Zhenhu He[1], Huiling Li[1], Wei Nie[1], Yin Liu[1], Jing Zhao[1], Muqing Luo[1], Xuanhui Liu[7], Pengfei Rong [1,8✉] & Wei Wang [1,8✉]

The outbreak of coronavirus disease 2019 (COVID-19) has rapidly spread to become a worldwide emergency. Early identification of patients at risk of progression may facilitate more individually aligned treatment plans and optimized utilization of medical resource. Here we conducted a multicenter retrospective study involving patients with moderate COVID-19 pneumonia to investigate the utility of chest computed tomography (CT) and clinical characteristics to risk-stratify the patients. Our results show that CT severity score is associated with inflammatory levels and that older age, higher neutrophil-to-lymphocyte ratio (NLR), and CT severity score on admission are independent risk factors for short-term progression. The nomogram based on these risk factors shows good calibration and discrimination in the derivation and validation cohorts. These findings have implications for predicting the progression risk of COVID-19 pneumonia patients at the time of admission. CT examination may help risk-stratification and guide the timing of admission.

[1] Department of Radiology, Third Xiangya Hospital, Central South University, Changsha, Hunan, China. [2] Department of Radiology, First Hospital of Changsha, Changsha, Hunan, China. [3] Changsha Public Health Treatment Center, Changsha, Hunan, China. [4] Department of Medical Imaging, First Hospital of Yueyang, Yueyang, Hunan, China. [5] Department of Medical Imaging, Central Hospital of Shaoyang, Shaoyang, Hunan, China. [6] Westmead Clinical School, Faculty of Medicine and Health, University of Sydney, Sydney, Australia. [7] Second People's Hospital of Hunan, Changsha, Hunan, China. [8] Molecular Imaging Research Center, Central South University, Changsha, Hunan, China. ✉email: rongpengfei66@163.com; cjr.wangwei@vip.163.com

The outbreak of coronavirus disease 2019 (COVID-19), caused by severe acute respiratory syndrome coronavirus 2 (SARS-CoV-2), has rapidly spread to become a worldwide pandemic. Most COVID-19 patients have a mild clinical course, while a proportion of patients demonstrated rapid deterioration (particularly within 7–14 days) from the onset of symptoms into severe illness with or without acute respiratory distress syndrome (ARDS)[1,2]. There is no specific anti-coronavirus treatment for severe patients at present, and whether remdesivir is associated with significant clinical benefits for severe COVID-19 still requires further confirmation[3,4]. These patients have poor survival and often require intensive medical resource utilization, and their case fatality rate is about 20 times higher than that of nonsevere patients[5,6]. Thus, early identification of patients at risk of severe complications of COVID-19 is of clinical importance. Several studies reported that the prevalence of severe COVID-19 ranged from 15.7 to 26.1% among patients admitted to hospital and these cases were often associated with abnormal chest computed tomography (CT) findings and clinical laboratory data[6–8]. Guan et al. indicated that severe COVID-19 patients were more likely to show ground-glass opacity (GGO), local or bilateral patchy shadowing, and interstitial abnormalities on CT[8]. This likely reflects the clinical progression of disease but also offers an opportunity to investigate the clinical utility of chest CT as a predictive tool to risk-stratify the patients. Furthermore, the predictive value of chest CT for the prognosis of COVID-19 patients is warranted to assist the effective treatment and control of disease spread. Previous study suggested that higher CT lung scores correlated with poor prognosis in patients with Middle East Respiratory Syndrome (MERS)[9]. Chest CT has been proposed as an ancillary approach for screening individuals with suspected COVID-19 pneumonia during the epidemic period and monitoring treatment response according to the dynamic radiological changes[10–12]. Therefore, we retrospectively enrolled patients with moderate COVID-19 pneumonia on admission from multiple hospitals and observed for at least 14 days to explore the early CT and clinical risk factors for progression to severe COVID-19 pneumonia, and constructed a nomogram based on the independent factors. Meanwhile, we also compared the clinical and CT characteristics in patients with different exposure history or period from symptom onset to admission to provide a deep understanding of the relationship among CT findings, epidemiological features, and inflammation.

Here, we show a nomogram incorporating age, neutrophil-to-lymphocyte ratio (NLR), and CT severity score on admission with good performance in the prediction of short-term disease progression in hospitalized patients with moderate COVID-19 pneumonia, which have implications for early predicting the progression risk and guiding individually aligned treatment plans among COVID-19 pneumonia patients.

## Results

**Patient characteristics**. A total of 298 patients with COVID-19 were identified according to the inclusion criteria, of which 51 patients were excluded for having: (1) negative CT findings or severe/critical COVID-19 on admission ($n = 39$); (2) age younger than 18 years old ($n = 12$). Finally, 247 patients were included in our study, consisting of 141 patients in the derivation cohort and 106 patients in the validation cohort (Fig. 1). The clinical characteristics of included patients in the derivation and validation cohorts are presented in Table 1. Among them, 72 (51.1%) and 54 (50.9%) patients were male, with a median age of 44 years and 46 years in the derivation and validation cohorts, respectively. There were no significant differences in the clinical characteristics between the two cohorts (Supplementary Data 1). During the

hospitalization, 15/141 (10.6%) and 10/106 (9.4%) patients progressed to severe pneumonia in the derivation and validation cohorts, and 6/15 (40.0%) and 5/10 (50.0%) severe cases further deteriorated critical illness. Besides, 245 (99.2%) patients of the included patients had recovered and discharged at the time of analysis.

**Comparison between stable and progressive patients**. The clinical and CT characteristics of patients who progressed to severe COVID-19 (progressive group) and patients who did not (stable group) in the derivation cohort are presented in Table 2. Compared with those in the stable group, patients in the progressive group were significantly older (median age, 58 vs. 41 years old, $P = 0.001$) and more likely to have underlying hypertension ($P = 0.004$). But no significant difference was found in gender, exposure history, smoking history, and other co-morbidities including diabetes, chronic obstructive pulmonary disease (COPD), cardiovascular disease, cerebrovascular disease, and chronic hepatitis B infection between the two groups. The main clinical symptoms between the two groups were not statistically different, while slightly more patients manifested anorexia ($P = 0.088$), diarrhea ($P = 0.065$), and shortness of breath ($P = 0.088$) in the progressive group. There was no significant difference in percutaneous oxygen saturation on admission between the two groups ($P = 0.110$). Patients in the progressive group had lower baseline lymphocyte count and albumin, and higher NLR, aspartate aminotransferase, lactic dehydrogenase, and C-reactive protein (all $P < 0.05$). The major CT features of COVID-19 pneumonia patients were bilateral, peripheral or mixed distributed GGO, consolidation, and GGO with consolidation (Fig. 2a–f). Patients in the progressive group had more lobes and segments involved, with a higher proportion of crazy-paving sign and higher CT severity score (Supplementary Fig. 1a) compared with those in the stable group (all $P < 0.05$). However, no significant difference was found in hospital length of stay ($P = 0.398$) and duration of viral shedding after illness onset ($P = 0.087$) between the two groups.

**Logistic regression analysis and nomogram establishment**. Multivariate logistic regression analysis showed that age (odds ratio [OR] and 95% confidence interval [CI], 1.06 [1.01–1.12]; $P = 0.028$), baseline NLR (OR and 95% CI, 1.74 [1.13–2.70]; $P = 0.011$), and CT severity score (OR with 95% CI 1.19 [1.01–1.41]; $P = 0.043$) were independent predictors for progression to severe COVID-19 pneumonia in the derivation cohort (Table 3). A nomogram incorporating these three predictors was then constructed (Fig. 3a). The calibration curve of the nomogram (Fig. 3b) and a nonsignificant Hosmer–Lemeshow test statistic ($P = 0.378$) showed good calibration in the derivation cohort. The favorable calibration of the nomogram was confirmed with the validation cohort (Fig. 3c), with a nonsignificant Hosmer–Lemeshow test statistic ($P = 0.791$). The area under the receiver operating characteristic curve (AUC) of the nomogram in the derivation and validation cohorts was 0.867 (95% CI, 0.770–0.963; Fig. 3d) and 0.898 (95% CI, 0.812–0.984; Fig. 3e), respectively, which revealed good discrimination. Therefore, our nomogram performed well in both the derivation and validation cohorts. The nomogram has been deployed in an online risk calculator, which is freely available at https://xy3yyfskfzc.shinyapps.io/DynNomapp2/. The decision curve analysis (DCA) for the nomogram in the derivation cohort is presented in Supplementary Fig. 2. The DCA indicated that when the threshold probability for a doctor or a patient was within a range from 0.02 to 0.92, the nomogram added more net benefit than the "treat all" or "treat none" strategies.

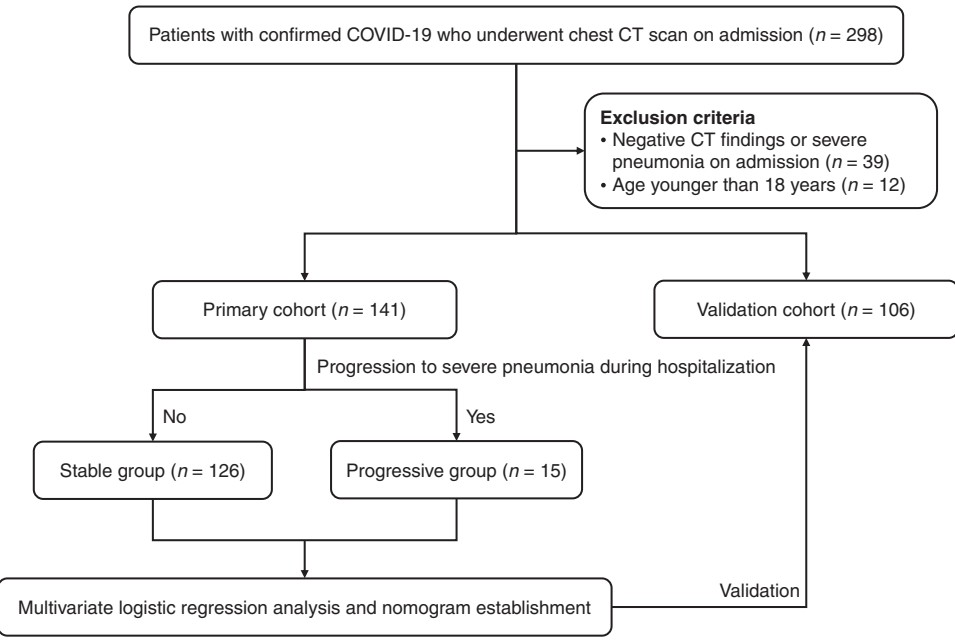

**Fig. 1 Study workflow.** The flow diagram shows the study population enrollment and observation period.

**Association of CT characteristics with inflammatory indexes.** In the derivation cohort, correlation analysis showed that there were significantly positive associations between baseline CT severity score and inflammatory indexes (neutrophil count, lactic dehydrogenase, and C-reactive protein) on admission (Fig. 4a). Furthermore, for 134 patients who had available inflammatory indexes results on day 3 after admission in the derivation cohort, baseline CT severity score was not only positively associated with inflammatory indexes (white blood cell count, neutrophil count, lactic dehydrogenase, and C-reactive protein), but also negatively associated with lymphocyte count on day 3 after admission (Fig. 4b), which indicated the potential of CT severity score for early predicting lymphopenia.

**Comparison between patients with and without bacterial co-infection.** Bacterial coinfection may complicate the disease course, which needs to be treated using antibiotics. We further investigated the prevalence of bacterial coinfection in COVID-19 pneumonia patients and its association with clinical and CT characteristics in the derivation cohort (Supplementary Table 1). 17/141 (12.1%) developed bacterial coinfection during hospitalization. Patients who developed bacterial coinfection were significantly older (median age, 56 vs. 41 years old, $P = 0.031$), more likely to have underlying hypertension ($P = 0.012$), and less likely to show consolidation on CT at the time of admission ($P = 0.012$) while had comparable CT severity score ($P = 0.817$; Supplementary Fig. 1b) compared those without bacterial coinfection. Furthermore, patients with bacterial coinfection had significantly longer hospital length of stay and duration of viral shedding after illness onset (both $P < 0.001$).

**Comparison between other subgroups.** There was no significant difference in the clinical and CT characteristics between patients with and without exposure history in Wuhan within 2 weeks before illness onset in the derivation cohort (Supplementary Table 2), who had similar age, NLR level, and CT severity score (Supplementary Fig. 1c). In addition, the median period from symptom onset to admission in the derivation cohort was 4 days. For patients who were admitted more than 4 days from symptom onset, more lobes or segments involved, and higher CT severity

score (Supplementary Fig. 1d) were found, while there was no significant difference in age and inflammatory indexes between patients with different period (≤4 days vs. >4 days) from symptom onset to admission in the derivation cohort (Table 4).

## Discussion

In this study, we retrospectively assessed the clinical and CT characteristics of COVID-19 pneumonia patients from multiple hospitals and identified the baseline risk factors for clinical progression. Our results indicated that CT severity score was associated with inflammatory levels, and older age, higher NLR and CT severity score on admission were independent predictors for progression to severe COVID-19 pneumonia. The nomogram based on these risk factors showed good calibration and discrimination in the derivation and validation cohorts. In addition, patients who were admitted longer from symptom onset had more severe lung involvement.

With the rapid increase of newly confirmed and severe cases, the management of severe patients become a challenging issue of the COVID-19 outbreak. Timely identification of patients with high risk to develop ARDS or multiorgan failure and risk stratification management might be helpful for more individually aligned treatment plans, optimized utilization of medical resource, and preventing further deterioration. In our cohorts, the prevalence of severe COVID-19 pneumonia was about 10%, which was lower than that in some large-scale reports[6,8]. This might be explained by the inclusion of only moderate patients on admission. Besides, the patients in our cohort were younger compared with those in Wuhan, which may be due to the fact that most of the patients with exposure history in Wuhan were young or middle-aged individuals working in Wuhan[13]. We found that progressive patients were more likely to be older and had underlying hypertension compared with stable patients. These data were in agreement with recent reports, which suggested that age and hypertension may be risk factors for progression in COVID-19 patients[14–16].

COVID-19 pneumonia patients who progressed in disease severity had lower baseline lymphocyte count, and higher NLR, lactic dehydrogenase, and C-reactive protein. SARSCoV-2 virus might act on lymphocytes as does SARSCoV, which induces a

## Table 1 Clinical characteristics of patients with COVID-19 pneumonia in the derivation and validation cohorts.

| Variables | Derivation (n = 141) | Validation (n = 106) | P value |
|---|---|---|---|
| Age (years) | 44 (34–55) | 46 (35–56) | 0.695 |
| Male gender | 72 (51.1%) | 54 (50.9%) | 0.985 |
| *Exposure history in Wuhan within 2 weeks* | | | 0.075 |
| Yes | 76 (53.9%) | 45 (42.5%) | |
| No | 65 (46.1%) | 61 (57.6%) | |
| Smoking history | 7 (5.0%) | 7 (6.6%) | 0.581 |
| *Comorbidities* | | | |
| Any | 33 (23.4%) | 21 (19.8%) | 0.499 |
| Diabetes | 8 (5.7%) | 6 (5.7%) | 0.996 |
| Hypertension | 21 (14.9%) | 10 (9.4%) | 0.200 |
| Cardiovascular disease | 3 (2.1%) | 2 (1.9%) | 0.894 |
| COPD | 4 (2.8%) | 3 (2.8%) | 0.997 |
| Cerebrovascular disease | 1 (0.7%) | 0 (0) | 0.385 |
| Hepatitis B infection | 4 (2.8%) | 2 (1.9%) | 0.703 |
| *Laboratory findings* | | | |
| Lymphocyte count (×10$^9$/L) | 1.1 (0.8–1.5) | 1.1 (0.9–1.6) | 0.350 |
| NLR | 2.6 (1.9–3.7) | 2.7 (1.7–3.7) | 0.769 |
| Aspartate aminotransferase (U/L) | 24.0 (19.9–30.7) | 25.4 (19.9–34.0) | 0.272 |
| Albumin (g/L) | 37.1 (34.8–40.1) | 37.9 (34.9–40.3) | 0.348 |
| Lactic dehydrogenase (U/L) | 175.9 (138.9–221.9) | 183.4 (145.1–247.1) | 0.154 |
| C-reactive protein (mg/L) | 17.4 (7.4–38.2) | 16.9 (2.6–39.9) | 0.191 |
| *Radiological findings* | | | |
| Bilateral involvement | 123 (87.2%) | 94 (88.7%) | 0.731 |
| CT severity score | 6 (4–10) | 7 (4–10) | 0.149 |
| *Clinical outcomes* | | | |
| Severe pneumonia | 15 (10.6%) | 10 (9.4%) | 0.756 |
| Requiring mechanical ventilation | 6 (4.3%) | 5 (4.7%) | 0.862 |
| ICU admission | 4 (10.6%) | 4 (3.8%) | 0.728 |
| Death | 1 (0.7%) | 1(0.9%) | 0.839 |

Data are presented as median (IQR) or n (%). Differences between groups are analyzed using Student's *t*-test or Mann–Whitney *U*-test for continuous variables and Chi-square test or Fisher's exact test for categorical variables. Two-sided *P*-values are reported.
*COPD* chronic obstructive pulmonary disease, *COVID-19* coronavirus disease 2019, *CT* computed tomography, *ICU* intensive care unit, *IQR* interquartile range, *NLR* neutrophil-to-lymphocyte ratio.

## Table 2 Clinical and CT characteristics between the stable and progressive patients in the derivation cohort.

| Variables | Stable (n = 126) | Progressive (n = 15) | P value |
|---|---|---|---|
| Age (years) | 41 (33–52) | 58 (44–66) | 0.001 |
| Male (gender) | 65 (51.6%) | 7 (46.7%) | 0.719 |
| *Exposure history in Wuhan within 2 weeks* | | | 0.616 |
| Yes | 67 (53.2%) | 9 (60.0%) | |
| No | 59 (46.8%) | 6 (40.0%) | |
| Smoking history | 7 (5.6%) | 0 (0) | 0.349 |
| *Comorbidities* | | | |
| Any | 26 (20.6%) | 7 (46.7%) | 0.024 |
| Diabetes | 6 (4.8%) | 2 (13.3%) | 0.175 |
| Hypertension | 15 (11.9%) | 6 (40.0%) | 0.004 |
| Cardiovascular disease | 2 (1.6%) | 1 (6.7%) | 0.288 |
| COPD | 2 (1.6%) | 2 (13.3%) | 0.056 |
| Cerebrovascular disease | 1 (0.8%) | 0 (0) | 0.729 |
| Hepatitis B infection | 4 (3.2%) | 0 (0) | 0.484 |
| *Signs and symptoms* | | | |
| Fever | 92 (73.0%) | 13 (86.7%) | 0.252 |
| Cough | 66 (52.4%) | 8 (53.3%) | 0.944 |
| Sputum production | 14 (11.1%) | 2 (13.3%) | 0.798 |
| Fatigue or myalgia | 28 (22.2%) | 3 (20.0%) | 0.844 |
| Anorexia | 3 (2.4%) | 2 (13.3%) | 0.088 |
| Diarrhea | 4 (3.2%) | 2 (13.3%) | 0.065 |
| Shortness of breath | 3 (2.4%) | 2 (13.3%) | 0.088 |
| Percutaneous oxygen saturation (%) | 97.5 (96.1–98.6) | 95.6 (94.5–98.5) | 0.110 |
| *Laboratory findings* | | | |
| Platelet count (×10$^9$/L) | 170.5 (137.8–224.0) | 148.0 (121.0–204.0) | 0.192 |
| White blood cell count (×10$^9$/L) | 4.4 (3.4–5.2) | 4.6 (3.3–5.7) | 0.683 |
| Neutrophil count (×10$^9$/L) | 2.8 (2.1–3.6) | 3.2 (2.4–4.4) | 0.237 |
| Lymphocyte count (×10$^9$/L) | 1.1 (0.9–1.5) | 0.7 (0.5–1.3) | 0.002 |
| NLR | 2.5 (1.8–3.4) | 4.8 (3.1–5.1) | <0.001 |
| Alanine aminotransferase (U/L) | 20.0 (14.5–28.4) | 19.0 (14.2–31.4) | 0.794 |
| Aspartate aminotransferase (U/L) | 23.2 (19.6–28.9) | 30.4 (25.2–37.4) | 0.013 |
| Total bilirubin (μmol/L) | 11.3 (8.9–15.7) | 10.6 (8.8–13.8) | 0.339 |
| Albumin (g/L) | 37.5 (35.3–40.2) | 35.0 (31.7–37.3) | 0.008 |
| Creatinine (μmol/L) | 49.8 (40.1–60.3) | 52.3 (33.6–63.9) | 0.683 |
| Creatine kinase (U/L) | 69.6 (40.5–122.3) | 92.0 (57.5–386.0) | 0.120 |
| Lactic dehydrogenase (U/L) | 168.9 (134.9–217.3) | 197.4 (182.8–276.9) | 0.014 |
| C-reactive protein (mg/L) | 15.7 (6.8–36.8) | 36.9 (27.2–58.7) | 0.001 |
| *CT features* | | | |
| Number of lobes involved | | | 0.013 |
| One lobe | 12 (9.5%) | 2 (13.3%) | |
| Two lobes | 25 (19.9%) | 0 (0) | |
| Three lobes | 15 (11.9%) | 0 (0) | |
| Four lobes | 28 (22.2%) | 1 (6.7%) | |
| Five lobes | 46 (36.5%) | 12 (80.0%) | |
| Number of segments involved | 9 (5–12) | 12 (10–15) | 0.007 |
| Bilateral involvement | 110 (87.3%) | 13 (86.7%) | 0.944 |
| Distribution pattern | | | 0.116 |
| Peripheral | 67 (53.2%) | 4 (26.7%) | |

cytokine storm and triggers a series of immune responses[17]. Some studies suggested that the decrease of peripheral T lymphocyte count attributes to the inflammatory cytokine milieu and T cell recruitment to sites of infection, and reduced but hyperactivated or exhausted peripheral T cells were more frequently found in severe cases[18,19]. Lymphopenia has been confirmed as a potential factor associated with disease severity and mortality in COVID-19[13]. Thus, damage to lymphocytes and consequently immunologic abnormality might be an important factor leading to exacerbations of patients. The uncontrolled inflammatory response could also stimulate the production of neutrophils apart from speeding up the apoptosis of lymphocytes[20]. NLR, a simple biomarker to assess the systemic inflammatory status, is widely used for the prediction of prognosis of patients with pneumonia[21,22]. Increased NLR, resulting from decreased lymphocyte count and/or elevated neutrophil count, represents damaged lymphocyte function and/or increased inflammatory level and risk of bacterial infection. In addition, C-reactive protein is another serum maker produced by the liver in response to

**Table 2 (continued)**

| Variables | Stable (n = 126) | Progressive (n = 15) | P value |
|---|---|---|---|
| Central | 2 (1.6%) | 0 (0) | |
| Mixed | 57 (45.2%) | 11 (73.3%) | |
| GGO | 120 (88.9%) | 15 (100%) | 0.388 |
| Consolidation | 107 (84.9%) | 13 (86.7%) | 0.858 |
| GGO with consolidation | 100 (79.4%) | 13 (86.7%) | 0.503 |
| Crazy-paving | 34 (27.0%) | 8 (53.3%) | 0.035 |
| Air bronchogram | 71 (56.4%) | 11 (73.3%) | 0.207 |
| Discrete nodules | 10 (7.9%) | 1 (6.7%) | 0.862 |
| Lymphadenopathy | 5 (4.0%) | 1 (6.7%) | 0.625 |
| Pleural effusion | 4 (3.2%) | 0 (0) | 0.484 |
| CT severity score | 6 (4–9) | 10 (7–15) | 0.001 |
| Hospital length of stay (days) | 21 (16–28) | 22 (18–35) | 0.398 |
| Duration of viral shedding after illness onset (days) | 14 (10–24) | 18 (13–31) | 0.087 |

Data are presented as median (IQR) or n (%). Differences between groups are analyzed using Student's t-test or Mann–Whitney U-test for continuous variables and Chi-square test or Fisher's exact test for categorical variables. Two-sided P values are reported.
*COPD* chronic obstructive pulmonary disease, *CT* computed tomography, *GGO* ground-glass opacities, *IQR* interquartile range, *NLR* neutrophil-to-lymphocyte ratio.

inflammation. Liu et al. reported that C-reactive protein might be predictive of disease severity in COVID-19 patients[23]. Thus, our results suggested that patients with higher inflammatory levels on admission had higher risk to develop severe COVID-19.

To explore the predictive value of chest CT for progression, we compared the difference of CT characteristics in the stable and progressive patients and found that progressive patients had higher CT severity score. CT severity score is used to semi-quantitatively estimate the pulmonary involvement, which is associated with both the number of involved lobes and extent of lesions[24]. In support of our findings, a previous report regarding MERS showed the predictive value of CT severity score for prognosis and short-term mortality[9]. Furthermore, a higher proportion of progressive patients showed crazy-paving sign which reflects interstitial thickening[25]. The binding of SARSCoV-2 spike protein to the receptor angiotensin-converting enzyme II (ACE2) contributes to the downregulation of ACE2, increased pulmonary capillary permeability, and diffuse alveolar damage[26–28]. In patients with SARS, mixed and predominant reticular patterns were also noted from the second week[29,30]. Therefore, we speculated that the involvement of the interstitial vascular endothelial cells results in interlobular and intralobular septal thickening, which may be associated with the disease severity.

Our results further revealed that age, NLR, and CT severity score on admission were significant predictors for progression in moderate COVID-19 pneumonia patients. The predictive value of age and NLR has been reported in recent studies[14,16,31]. Previous study showed that the MuLBSTA score could early warn the mortality of viral pneumonia, which included lymphopenia and multilobe infiltration[32]. Our findings were consistent with theirs but more quantitatively in terms of imaging evaluation of lung involvement. Furthermore, we constructed a nomogram based on the multivariate logistic regression model to provide an easy-to-use tool for clinicians in the prediction of severe pneumonia in COVID-19 patients, which showed good performance in both the derivation cohort and external validation cohort. Recently, Liang et al. proposed a clinical risk score incorporating ten clinical variables to predict the occurrence of critical illness in hospitalized patients with COVID-19[33]. Their risk score included dichotomous chest X-ray abnormality instead of the severity of abnormality on CT. In contrast to their study, we adopted a quantitative CT severity score to accurately assess the degree of lung injury and aimed to early predict the in-hospital progression risk within 14 days in patients with moderate COVID-19 pneumonia on admission. Furthermore, the prediction model established in our study was simpler with only three easily accessible variables compared with theirs. Like SARS and MERS, some COVID-19 pneumonia patients progressed rapidly at about 7–14 days after onset likely due to the cytokine storm in the body as evidenced by increased plasma proinflammatory cytokines[1,17,34]. The results revealed the significant association between baseline CT severity score and inflammatory markers, particularly baseline CT severity score and lymphocyte count at day 3 after admission, which implied the potential value of chest CT on admission to estimate pulmonary inflammation or lung damage and to early predict lymphopenia.

Patients with bacterial coinfection during hospitalization were older and more likely to have underlying hypertension than those without, which suggested that age and hypertension may be risk factors for concomitant bacterial infection. In addition, consolidation on baseline chest CT was less likely found in those with bacterial coinfection, which may be due to the weak antiviral immune response at the early stage of COVID-19 pneumonia in these older individuals with existing comorbidities[35]. The imaging findings may help physicians to identify the patients with higher risk of bacterial coinfection and those who need prophylactic antibiotic therapy to shorten hospital length of stay and duration of viral shedding. We also acknowledged that our findings were limited by the relatively small sample size, which should be interpreted with caution by clinicans and further confirmed by larger samples. Besides, patients who were admitted more than 4 days after symptom onset had higher CT severity scores, which likely attributed to the lung involvement progression as disease course extends[36]. CT examination may be important in guiding the rational timing of admission for the individual management of COVID-19 pneumonia patients.

There were some limitations in our study. First, our study was retrospectively conducted, and the distribution of patients was imbalanced with only about 10% of cases developing severe pneumonia. Second, adjuvant treatment during hospitalization have not yet been analyzed, and multiple inflammatory cytokines were not available in this study. More comprehensive investigation of the relationship between CT characteristics and cytokine storm induced by COVID-19 needs to be performed. Third, CT is not used widely outside China for patients with COVID-19. Though the management guidelines in China recommend chest CT as a routine examination for COVID-19 pneumonia, the American College of Radiology advocates that CT should not be used as a first-line test to diagnose COVID-19 and should be used sparingly and reserved for hospitalized, symptomatic patients with specific clinical indications, which may limit the broad applicability of our findings[37].

In conclusion, our results indicated that older age, higher NLR, and CT severity score on admission were independent risk factors for clinical progression in moderate COVID-19 pneumonia patients, and the nomogram based on the three risk factors showed favorable predictive accuracy in the derivation and validation cohorts. Chest CT has the potential to early predict the risk of progression and reflect disease severity as well, which may also help guide the timing of admission for COVID-19 pneumonia patients.

## Methods

**Patients**. Our study was retrospectively conducted in compliance with the Health Insurance Portability and Accountability Act. We were authorized by the Hunan Provincial Health Commission in the collection of clinical and radiological data under anonymization according to the standard of care, which required the

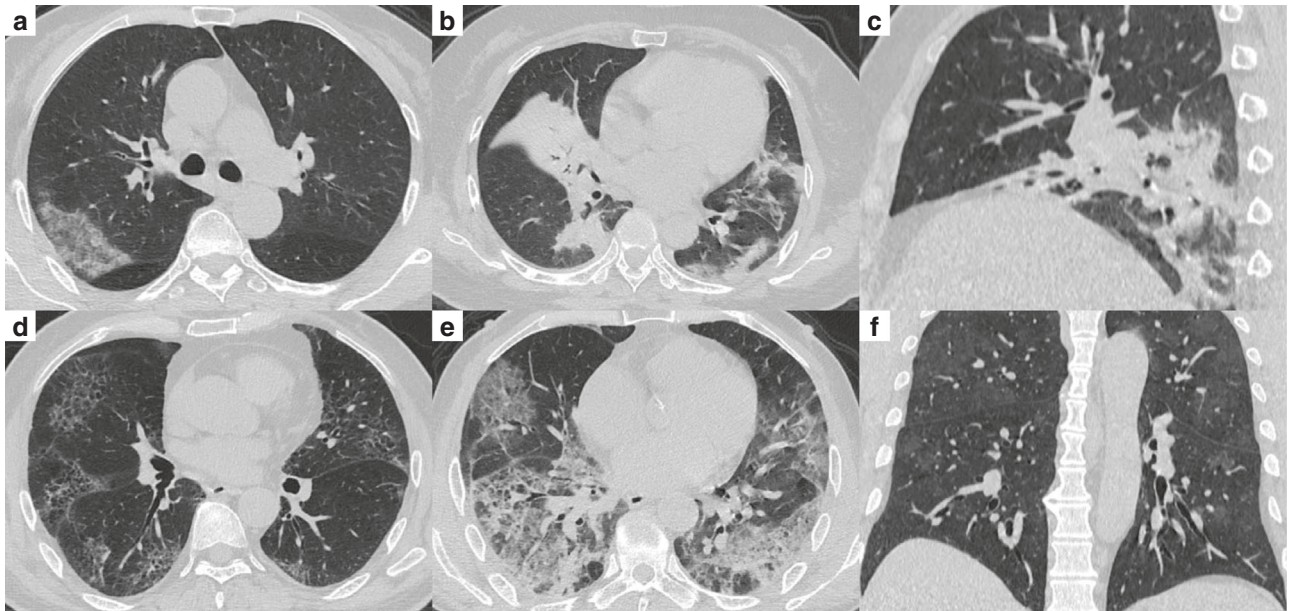

**Fig. 2 Representative chest CT images of patients with COVID-19 pneumonia. a** Subpleural patchy areas of GGO with crazy-paving sign in the right middle lobe. **b** Multiple patchy areas of consolidation in the right middle lobe, left upper lobe, and bilateral lower lobes and air bronchogram in the right middle lobe. **c** Multiple patchy areas of organizing pneumonia in the right middle and lower lobes on the sagittal image with CT severity score of 9 for the right lung. **d** Bilateral and peripheral multiple patchy areas of GGO with reticular and intralobular septal thickening. **e** Multiple mixed distributed pure GGO, GGO with consolidation, and interlobular septal thickening in bilateral lungs. **f** Bilateral multiple patchy and thin areas of GGO in the posterior parts of the lungs.

### Table 3 Risk factors for progression to severe COVID-19 pneumonia in the derivation cohort.

| Variables | OR (95% CI) | P value | OR (95% CI) | P value |
|---|---|---|---|---|
| Age | 1.09 (1.04–1.14) | 0.001 | 1.06 (1.01–1.12) | 0.028 |
| Hypertension | 4.93 (1.54–15.82) | 0.007 | | 0.676 |
| NLR | 2.13 (1.43–3.18) | <0.001 | 1.74 (1.13–2.70) | 0.012 |
| Aspartate aminotransferase | 1.04 (1.00–1.09) | 0.070 | | 0.682 |
| Albumin | 0.82 (0.71–0.96) | 0.011 | | 0.668 |
| Lactic dehydrogenase | 1.01 (1.00–1.02) | 0.006 | | 0.661 |
| C-reactive protein | 1.03 (1.01–1.06) | 0.004 | | 0.471 |
| Number of segments involved | 1.18 (1.04–1.35) | 0.013 | | 0.488 |
| Crazy-paving | 3.09 (1.04–9.18) | 0.042 | | 0.821 |
| CT severity score | 1.32 (1.14–1.54) | 0.001 | 1.19 (1.01–1.41) | 0.043 |

Univariate and multivariate logistic regression analyses are performed and the corresponding ORs are reported.
CI confidence interval, CT computed tomography, COVID-19 coronavirus disease 2019, NLR neutrophil-to-lymphocyte ratio, OR odds ratio.

implementation to meet the criteria of the Helsinki declaration and follow all relevant regulations regarding the use of human study participants. The permission of the Institutional Review Board of Third Xiangya Hospital was obtained with waiver of informed consent. None of the patients in this study were reported in prior publications regarding their clinical characteristics.

Health records were reviewed for patients who were treated at Third Xiangya Hospital, Changsha Public Health Treatment Center, Second People's Hospital of Hunan, First Hospital of Yueyang, and Central Hospital of Shaoyang between January 17, 2020 and February 1, 2020. Patients were included in the study if they satisfied the following criteria: (1) confirmed COVID-19; (2) available chest CT scan on admission. The diagnosis of COVID-19 was established based on the World Health Organization interim guidance, and a confirmed case was defined as a positive result to high-throughput sequencing or real-time reverse transcription-polymerase chain reaction (RT-PCR) assay of SARS-CoV-2 for nasal and pharyngeal swab specimens. Patients were observed for at least 14 days from admission to determine whether they exacerbated to severe pneumonia or not. We divided the patients into two independent cohorts: patients treated at Third Xiangya Hospital, Changsha Public Health Treatment Center, and Second People's Hospital of Hunan constituted the derivation cohort, whereas patients treated at First Hospital of Yueyang and Central Hospital of Shaoyang constituted the validation cohort.

**Clinical data collection.** Demographic information (age, gender), exposure history, smoking history, comorbidities (including diabetes, hypertension, cardio-vascular disease, COPD, cerebrovascular disease, and hepatitis B infection), clinical symptoms and signs (including percutaneous oxygen saturation), and laboratory data (platelet count, white blood cell count, neutrophil count, lymphocyte count, alanine aminotransferase, aspartate aminotransferase, total bilirubin, albumin, creatinine, creatine kinase, lactic dehydrogenase, and C-reactive protein) were obtained with data collection forms from electronic medical records. Exposure history was defined as exposure in Wuhan within 2 weeks before illness onset or exposure to local people with confirmed SARS-CoV-2 infection. The clinical classification of COVID-19 pneumonia is as follows: (1) moderate type, patients with fever, respiratory tract symptoms, and radiological evidence of confirmed pneumonia. (2) severe type, patients with one of the following: (a) respiratory distress (respiratory rate ≥ 30 beats/min); (b) hypoxia (oxygen saturation ≤ 93% in the resting state); (c) hypoxemia (arterial blood oxygen partial pressure/oxygen concentration ≤ 300 mmHg). (3) critical type, patients with one of the following: (a) respiratory failure requiring mechanical ventilation; (b) shock; (c) intensive care unit (ICU) admission is required for combined other organs failure. The primary endpoint of this study was the development of severe COVID-19 pneumonia by February 15, 2020, and other clinical outcomes including bacterial co-infection during hospitalization, requiring mechanical ventilation, ICU admission, discharge,

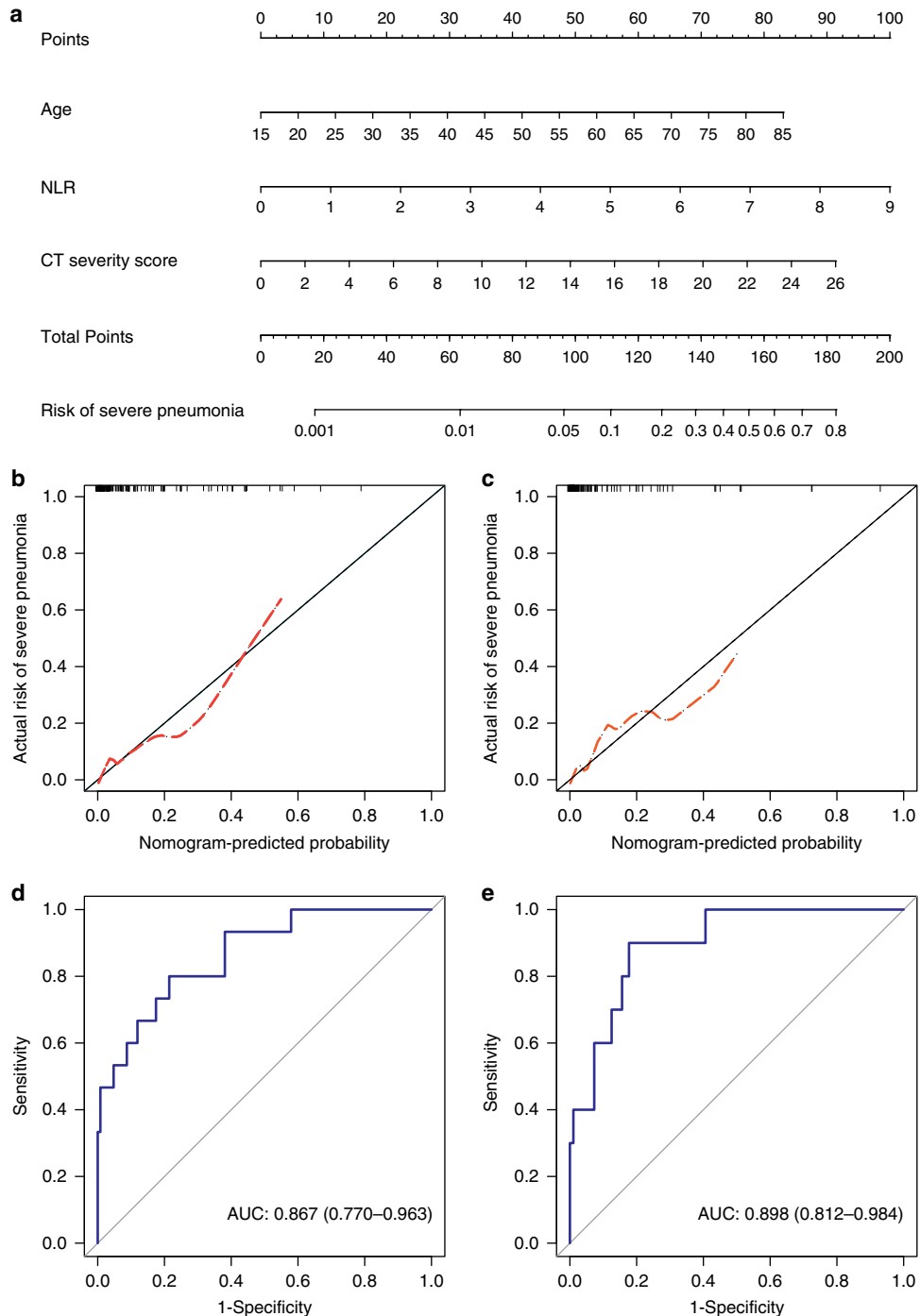

**Fig. 3 Development and performance of nomogram. a** A nomogram for the prediction of developing severe COVID-19 pneumonia. Calibration curves of the nomogram in the derivation (**b**) and validation (**c**) cohorts, respectively, which depict the calibration of the nomogram in terms of the agreement between the predicted risk of severe COVID-19 pneumonia and observed outcomes. The 45° blue line represents a perfect prediction, and the dotted red lines represent the predictive performance of the nomogram. The closer the dotted red line fit is to the ideal line, the better the predictive accuracy of the nomogram is. Plots show the ROC curves of the nomogram in the derivation (**d**) and validation (**e**) cohorts, respectively.

and death were also recorded. Bacterial coinfection was diagnosed when patients showed clinical symptoms, signs, or radiological evidence of nosocomial pneumonia or bacteremia and a positive bacterial culture test was obtained from lower respiratory tract specimens or blood samples after admission[1]. Hospital length of stay was calculated by subtracting day of admission from day of discharge. Duration of viral shedding after illness onset was considered as the number of days from symptom onset to persistent negative results on respiratory tract viral RT-PCR testing. All samples from the same patient were tested until two consecutive samples showed negative results, with the first negative result defining the duration of shedding. Several co-authors (L.L., W.Z., X.M., M.Y., T.L., and X.L.) only contributed to the collection of original data. The researchers who collected original data or extracted results of index tests were not involved in the final data summary and analysis.

**CT examination and image analysis**. All patients underwent chest CT examinations on admission. The images were reconstructed to 1.0-mm thickness for the transverse scans. Sagittal and coronal reconstructions with a 3.0-mm thickness were performed. All CT images were reviewed independently by two radiologists, each with over 10 years of experience in chest imaging. A third experienced radiologist was consulted if there was a disagreement in interpreting imaging results. The imaging features including GGO, consolidation, crazy-paving, and air

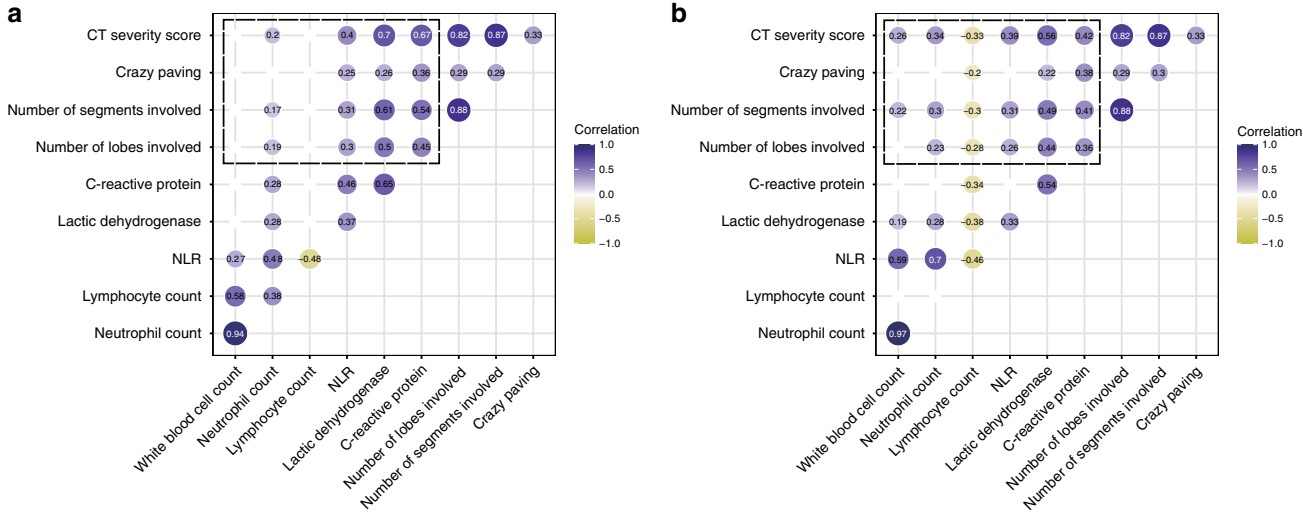

**Fig. 4 Correlation between CT characteristics and inflammatory indexes.** Heatmaps depict the correlations between the baseline CT characteristics and inflammatory indexes (within the blue dotted box) on admission (**a**) and on day 3 after admission (**b**) showing the correlation coefficients r with P < 0.05 of all pairs.

**Table 4 Clinical and CT characteristics of patients with COVID-19 in the derivation cohort according to the period from symptom onset to admission.**

| Variables | ≤4 days (n = 70) | >4 days (n = 71) | P value |
|---|---|---|---|
| Age (years) | 41 (31–53) | 46 (35–59) | 0.147 |
| Hypertension | 8 (11.4%) | 13 (18.3%) | 0.251 |
| Lymphocyte count (×10$^9$/L) | 1.1 (0.8–1.5) | 1.1 (0.9–1.4) | 0.944 |
| NLR | 2.7 (1.8–3.7) | 2.6 (1.9–3.9) | 0.867 |
| Aspartate aminotransferase (U/L) | 23.8 (19.8–30.4) | 24.0 (20.0–31.2) | 0.649 |
| Albumin (g/L) | 38.0 (35.3–40.6) | 36.4 (34.2–39.0) | 0.053 |
| Lactic dehydrogenase (U/L) | 173.3 (134.8–221.9) | 177.1 (141.9–232.1) | 0.598 |
| C-reactive protein (mg/L) | 18.1 (7.4–38.0) | 17.4 (6.8–38.7) | 0.918 |
| Number of lobes involved | 3 (2–5) | 4 (3–5) | 0.010 |
| Number of segments involved | 7 (3–12) | 10 (6–12) | 0.020 |
| Crazy-paving | 25 (35.7%) | 17 (23.9%) | 0.126 |
| CT severity score | 5 (3–10) | 7 (5–10) | 0.030 |

Data are presented as median (IQR) or n (%). Differences between groups are analyzed using Student's t-test or Mann–Whitney U-test for continuous variables and Chi-square test or Fisher's exact test for categorical variables. Two-sided P values are reported.
COVID-19 coronavirus disease 2019, CT computed tomography, IQR interquartile range, NLR neutrophil-to-lymphocyte ratio.

bronchogram were recorded[30,38]. GGO was defined as hazy increased lung attenuation with preservation of bronchial and vascular margins, whereas consolidation was defined as opacification with obscuration of margins of vessels and airway walls. Crazy-paving refers to the appearance of ground-glass opacity with superimposed interlobular septal thickening and intralobular septal thickening. The lesion distribution pattern, lobe and segment involvement were also assessed. The CT findings in the outer one third of the lung were defined as peripheral, and those in the inner two thirds of the lung were defined as central. Besides, the presence of discrete nodules, lymphadenopathy, and pleural effusion were recorded. Each of the five lung lobes was reviewed for opacification and consolidation. The lesions extent within each lung lobe was semiquantitatively evaluated by scoring from 0 to 5 based on the degree of involvement: score 0, none involvement; score 1, ≤5% involvement; score 2, 6–25% involvement; score 3, 26–50% involvement; score 4, 51–75% involvement; score 5, >75% involvement. The total score was calculated by summing up scores of all five lobes to provide a CT severity score ranging from 0 to 25[11].

**Statistical analysis**. Continuous variables are presented as the median and interquartile range, and categorical variables are presented as frequency and percentage. Differences between groups were analyzed using Student's t-test or Mann–Whitney U-test for continuous variables according to the normal distribution and Chi-square test or Fisher's exact test for categorical variables. Multivariate logistic regression with forward stepwise selection based on likelihood ratio was used to identify the risk factors for the development of severe COVID-19

in the derivation cohort, and a nomogram was then constructed by incorporating the independent factors. The calibration of the nomogram was assessed with a calibration curve and the Hosmer–Lemeshow test was performed to assess the goodness-of-fit. The discrimination performance of the nomogram was quantified using AUC. External validation of the nomogram was performed with the validation cohort. DCA was performed by calculating the net benefits of the nomogram for a range of threshold probabilities. The Spearman rank correlation was used in the correlation analysis. All clinical and imaging data items were entered into a worksheet in Microsoft Office Excel 2019. All statistical analyses were performed using SPSS statistics software (version 22.0, IBM Inc., Chicago, IL, USA) and R statistical software (version 3.6.1). A two-sided P value of less than 0.05 considered to be statistically significant.

**Reporting summary**. Further information on research design is available in the Nature Research Reporting Summary linked to this article.

## Data availability

The data supporting the main findings of this study are available from the corresponding authors upon reasonable request. A portion of data in this study is available within the Supplementary Information.

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

## Acknowledgements

We would like to show our respect for all the hospital staff for their hard work and efforts to combat the COVID-19. We thank Dr. Hongzhuan Tan, from Department of Epidemiology, Xiangya School of Public Health, Central South University for his statistical analysis discussion. We also thank Dr. Jianbin Tong, from Department of Anesthesiology, Dr. Quan Zhuang, and Dr. Qiquan Wan from Department of Transplantation, Third Xiangya Hospital, Central South University for their kind assistance in manuscript revision. This study was supported by the Wisdom Accumulation and Talent Cultivation Project of the Third Xiangya Hospital of Central South University.

## Author contributions

Z.F., Q.Y., P.R., and W.W. designed the study. S.Y., L.L., W.Z., X.M., M.Y., T.L., X.L., and P.R. collected the original clinical and imaging data. J.D., Z.Y., H.T., M.M., D.Y., Z.M., H.Z., Y.J., Z.H., H.L., W.N., J.Z., and M.L. extracted the main data. Z.F. conducted the statistical analysis. Z.F. and S.Y. wrote the original manuscript. Z.F., S.Y., J.L., Y.L., P.R., and W.W. revised the manuscript. P.R. and W.W. provided kind guidance for the research. All authors were involved in the drafting, review, and approval of the report and the decision to submit for publication.

## Competing interests

The authors declare no competing interests.
