## [Peer Review File · Nature Communications]

Reviewers' comments:

Reviewer #1 (Remarks to the Author):

This is another ms of COVID 19 from China. They studied the CT thorax characteristics of a cohort of patients hospitalized with COVID19 (141). They described the CT findings of this population and they compared the clinical severity with the CT findings. Logically they found CT differences between stable and non-stable patients. Patients with progression had lower lymphocyte counts lymphopenia at admission- The LR analyses found that base-line CT severity score and NLR ratio were factors associated were independent factors for progression

I have the following comments

1-This is an interesting clinical ms suggesting that CT scans can help in the prediction of progression of the disease.

-Please look at the following outcomes and not only progression

.ICU admission

.Mechanical ventilation

.LOS

.Crude and attributable mortality

2-Correlate the number of lymphocytes with the CT score severity

Might be very well that counting lymphocytes or the N/L ratio at the beginning and at day three could enough to predict progression

Please make correlations at day 1 and sequentially if you have the data

3-You do not mention bacterial co-infection

Please make the same analyses for those with co-infection

4-Please, report the Chest-X ray findings as well and if they have or not a good equivalence with CT findings

Based on these new analyses provide NEW recommendation for performing or not A CT scan in all hospitalized COVID patients

Reviewer #2 (Remarks to the Author):

GEN: The authors present an analysis of COVID-19 patients outside of the epicenter in Hubei province focused on CT findings important in prognosis. They confirm that CT severity score and neutrophil to lymphocyte ratio are associated with worsening of status. An interesting but unexplained finding is that the prognostic score did not work for imported cases.,.

MAJOR:

1. The authors combine their severe category with the critical category for analysis (line 141). This combination detracts severely from the value of the analysis. By their definitions, severe patients would likely be treated on a general medicine unit while critical would be ICU patients, with very different intervention needs. Any prognostic benefit of a score is therefore lost by this combination.

MINOR

1. The authors mention several times (e.g. lines 350-351, 361-363) that early admission is required to avoid severe lung damage. This is not proven by any data. The score is prognostic but the interventions available based on this score are unclear.

2.

Reviewer #3 (Remarks to the Author):

This is an interesting paper documenting that extent of disease on CT, and the neutrophil/lymphocyte ratio, are independent predictors of subsequent progression to severe disease. The number of patients is relatively large. The CT score is simple and would be easy to apply. The imaging illustrations are good.

Major limitations include the following:

Lack of validation in an independent population.

A major issue is that CT is not used widely outside China to evaluate COVID-19. The American College of Radiology has not endorsed the use of CT as a screening or first-line diagnosis tool for routine evaluation of COVID-19 infection (<https://www.acr.org/Advocacy-and-Economics/ACR-Position-Statements/Recommendations-for-Chest-Radiography-and-CT-for-Suspected-COVID19-Infection>).

This is primarily because of the need to deep clean and ventilate a CT room following any scan of patients with known or suspected COVID-19 infection. This issue limits the broad applicability of the findings, and should be addressed in the discussion as an important practical limitation to the use of CT.

Title

It is misleading to say in the Title and Abstract that the patients are from outside Wuhan, since about half of them came from or were exposed in Wuhan.

Abstract

Should specify the number of cases included. Should specify that the cohort was retrospective.

Introduction. It is not true that "Chest CT has been demonstrated to be an important approach for screening individuals with suspected COVID-19". Reference 10 shows that CT is relatively insensitive for early identification of COVID-19 infection. Reference 8 addresses only CT documentation of longitudinal course in 21 patients.

Methods

The authors should indicate if any of the patients reported here were included in prior publications. Were all patients directly admitted to the authors' hospital, or were some transferred from elsewhere? In particular, does the word "imported" mean that these patients were transferred from hospitals in Wuhan, or just that they were exposed in Wuhan? If they were transferred, the authors should provide the duration of admission from the date of admission in Wuhan. Please use a more specific word in place of "imported".

Results

The statement “52 patients of the included patients had discharged at enrollment.” Is unclear. Were these patients discharged without hospitalization?

Why were patients with normal CT findings or those with severe COVID excluded?

Smoking history seems remarkably low compared to the prevalence of smoking in China. Is this because of incomplete data collection? How did the authors handle cases where smoking history was unknown?

Likewise the age distribution appears younger than in Wuhan, particularly for the more severe cases. The authors should comment on this difference.

Did all patients know the date of exposure? This is surprising, as some patients may have been exposed to asymptomatic carriers.

Were other indices of disease severity measured? What about oxygen saturation, or oxygen requirement?

Table 1. What are the numbers provided? Some appear to be mean (or median?) with standard deviation, some appear to be mean with range.

Figure 3. Unclear what the shaded markings, and the white rectangles refer to. Please explain in figure legend. Please also include a marker of the mean and standard deviation.

Figure 4. I don't think that this conveys much useful information, since the correlations between imaging and nonimaging features

Point-by-point answers to the reviewers' comments.

Their comments are reproduced and our responses are given afterward in bold.

All changes in the main text are in red color.

Reviewers' comments:

Reviewer #1 (Remarks to the Author):

This is another ms of COVID 19 from China. They studied the CT thorax characteristics of a cohort of patients hospitalized with COVID19 (141). They described the CT findings of this population and they compared the clinical severity with the CT findings. Logically they found CT differences between stable and non-stable patients. Patients with progression had lower lymphocyte counts lymphopenia at admission- The LR analyses found that base-line CT severity score and NLR ratio were factors associated were independent factors for progression

I have the following comments

1-This is an interesting clinical ms suggesting that CT scans can help in the prediction of progression of the disease.

-Please look at the following outcomes and not only progression

.ICU admission

.Mechanical ventilation

.LOS

.Crude and attributable mortality

***Response:* Thank you for your recognition of the importance of our work, and we appreciate your insightful comments. In the revised manuscript, we supplemented the missing values of the laboratory test results in the derivation cohort, established a nomogram based on multivariate logistic regression model, and further validated the findings in an external cohort. According to the reviewer's suggestion, we have added other clinical outcomes including requiring mechanical ventilation, ICU admission, and death in both the derivation and validation cohorts (Table 1, Page 9). In addition, the hospital length of stay and duration of viral shedding after illness onset were also provided and compared between stable and progressive patients (Table 2, Page 11), and between patients with and without bacterial co-infection during hospitalization (Supplementary Table 1, Page 27).**

2-Correlate the number of lymphocytes with the CT score severity

Might be very well that counting lymphocytes or the N/L ratio at the beginning and at day three could be enough to predict progression

Please make correlations at day 1 and sequentially if you have the data

Response: Thank you for your insightful suggestions. In the derivation cohort (n = 141), 134 patients had available inflammatory indexes (white blood cell count, neutrophil count, lymphocyte count, NLR, lactic dehydrogenase, C-reactive protein) results on day 3 after admission. We performed the correlation analysis between the baseline CT severity score and these inflammatory indexes on day 3 after admission. Our results showed that the baseline CT severity score was negatively associated with lymphocyte count on day 3 after admission but not with that on admission, which indicated the potential of chest CT for early predicting lymphopenia (*Fig. 4b, Page 16*). This also has been indicated in the revised manuscript (*Line 258-267, Page 15*).

3-You do not mention bacterial co-infection

Please make the same analyses for those with co-infection

Response: Bacterial co-infection may complicate the COVID-19 disease course, which needs to be treated using antibiotics. We investigated the prevalence of bacterial co-infection during hospitalization in COVID-19 pneumonia patients and its association with clinical and CT characteristics in the derivation cohort (*Supplementary Table 1, Page 25-26*). The results showed that patients with bacterial co-infection were older and more likely to have underlying hypertension, and they were less likely to show consolidation in chest CT on admission compared those without bacterial co-infection. Besides, we found that patients with bacterial co-infection had longer hospital length of stay and duration of viral shedding after illness onset. These findings may be helpful for the use of prophylactic antibiotics in patients with COVID-19 pneumonia. This has been indicated in the revised manuscript (*Line 274-285, Page 16*).

4-Please, report the Chest-X ray findings as well and if they have or not a good equivalence with CT findings

Based on these new analyses provide NEW recommendation for performing or not A

CT scan in all hospitalized COVID patients

Response: Thank you for your comments. In the very early stage of the COVID-19 outbreak in China, chest X-ray was used to preliminarily screen suspected individuals. However, compared with chest high-resolution CT, the sensitivity of chest X-ray was insufficient and there was a high risk of missed diagnosis. Here we summarized the radiological findings of chest X-ray and CT scans in 37 patients with COVID-19 (Table 1 below), which was consistent with the report by Guan et al.¹ Since the missed diagnosis screened by chest X-ray may lead to delay confirmation and treatment, the management guidelines in China recommend chest CT as a routine screening approach for COVID-19 pneumonia. Chest X-ray can be used to monitor the progression of pulmonary involvement in patients with severe or critical cases. Besides, we also provided two representative cases in which chest X-ray showed no abnormalities but CT scan confirmed typical abnormalities (Figure 1, 2 below). Thus, based on our clinical experience and the findings in this study, chest CT examinations should be performed at the time of admission for the hospitalized COVID-19 pneumonia patients to evaluate the lung injury severity. Meanwhile, dedicated CT scanners and proper cleaning protocols should be recommended.

Table 1 Pair comparison between chest X-ray and CT in 37 patients with COVID-19

	All patients (n = 37)
Distinctive abnormalities on chest X-ray	Not available
Patchy shadowing	22/37 (59.5%)
Bundle-shaped or striped shadowing	5/37 (13.5%)
Bilateral involvement	17/37 (45.9%)
Distinctive abnormalities on chest CT	35/37 (94.6%)
Ground-glass opacity	31/37 (83.8%)
Consolidation	22/37 (59.5%)
Bilateral involvement	32/37 (86.5%)

Figure 1

Figure 2

Reviewer #2 (Remarks to the Author):

GEN: The authors present an analysis of COVID-19 patients outside of the epicenter in Hubei province focused on CT findings important in prognosis. They confirm that CT severity score and neutrophil to lymphocyte ratio are associated with worsening of status. An interesting but unexplained finding is that the prognostic score did not work for imported cases.,.

***Response:* Special thanks to you for your good comments. Our results showed that there was no difference in clinical and CT imaging characteristics between imported (patients with exposure history in Wuhan within 2 weeks before illness onset) and nonimported patients, and the exposure history was also not associated with progression to severe COVID-19 pneumonia. However, the sensitivity analysis in the original manuscript indicated that the risk factors did not work for imported cases, which may be attributed to the small sample size of severe cases. In order to provide reliable clinical findings to readers, we did not perform the sensitivity analysis according to the exposure history in the revised manuscript. Besides, we supplemented the missing values of the laboratory test results in the derivation cohort, established a nomogram based on the multivariate logistic regression model, and further validated the findings in an external cohort.**

MAJOR:

1. The authors combine their severe category with the critical category for analysis (line 141). This combination detracts severely from the value of the analysis. By their definitions, severe patients would likely be treated on a general medicine unit while critical would be ICU patients, with very different intervention needs. Any prognostic benefit of a score is therefore lost by this combination.

***Response:* As pointed out by the reviewer, there are different clinical characteristics and management protocols (such as mechanical ventilation, glucocorticoids, or circulatory support) between severe and critical COVID-19 patients. In this study, patients with moderate COVID-19 pneumonia on admission were included and observed for at least 14 days from admission. The primary endpoint of this study was the development of severe COVID-19 pneumonia by February 15, 2020, and other clinical outcomes including**

requiring mechanical ventilation, ICU admission, and death were also recorded. Our study was to explore the early clinical and imaging risk factors for progression to severe COVID-19 pneumonia. In the derivation and validation cohorts of this study, 15/141 (10.6%) and 10/106 (9.4%) cases progressed to severe pneumonia, and 6/15 (40.0%) and 5/10 (50.0%) severe cases further deteriorated critical illness with requiring mechanical ventilation or even ICU admission. The nomogram established in the study help to identify the COVID-19 pneumonia patients who are at high risk to develop severe pneumonia. According to the reviewer’s suggestion, we did not mention the

combination of severe and critical COVID-19 as a single category any more in the revised manuscript (*Line 111-122, Page 5*).

MINOR

1. The authors mention several times (e.g. lines 350-351, 361-363) that early admission is required to avoid severe lung damage. This is not proven by any data. The score is prognostic but the interventions available based on this score are unclear.

Response: Thank you for your insightful comments. Our results revealed that for patients who were admitted more than 4 days from symptom onset, more lobes or segments involved and higher CT severity score were found than those who were admitted less than 4 days from symptom onset (*Table 4, Page 17*), which indicated the potential association between early admission and lung damage extent for COVID-19 pneumonia patients. In the revised manuscript, we confirmed that older age, higher neutrophil-to-lymphocyte ratio (NLR) and CT severity score were independent predictors for progression to severe COVID-19. We agree with the reviewer that the interventions based on these risk factors are unavailable here. Our team focused on early risk factors for severe pneumonia in

the current study and is working on adjuvant therapy associated with preventing severe pneumonia, and the results will be shared in another report (submitted now).

Reviewer #3 (Remarks to the Author):

This is an interesting paper documenting that extent of disease on CT, and the neutrophil/lymphocyte ratio, are independent predictors of subsequent progression to severe disease. The number of patients is relatively large. The CT score is simple and would be easy to apply. The imaging illustrations are good.

Response: Special thanks to you for your good comments.

Major limitations include the following:

Lack of validation in an independent population.

Response: Thank you for the constructive suggestion. To validate the findings in the derivation cohort (from 3 hospitals in Changsha, Hunan), we enrolled another external validation cohort (from 2 hospitals Yueyang and Shaoyang, Hunan) using the same inclusion and exclusion criteria. We also supplemented the missing values of the laboratory test results in the derivation cohort and established a nomogram based on the results of multivariate logistic regression analysis. Finally, the nomogram showed good calibration and discrimination in the derivation and validation cohorts, which were provided in the revised manuscript (Page 7-15).

A major issue is that CT is not used widely outside China to evaluate COVID-19. The American College of Radiology has not endorsed the use of CT as a screening or first-line diagnosis tool for routine evaluation of COVID-19 infection (<https://www.acr.org/Advocacy-and-Economics/ACR-Position-Statements/Recommendations-for-Chest-Radiography-and-CT-for-Suspected-COVID19-Infection>). This is primarily because of the need to deep clean and ventilate a CT room following any scan of patients with known or suspected COVID-19 infection. This issue limits the broad applicability of the findings, and should be addressed in the discussion as an important practical limitation to the use of CT.

Response: As pointed out by the reviewer, the American College of Radiology recommends that CT should not be used to screen for or as a first-line test to diagnose COVID-19 and should be used sparingly and reserved for hospitalized, symptomatic patients with specific clinical indications. Thus, CT is indeed not

used widely outside China for patients with COVID-19. However, our preliminary clinical experience and the results in the current study indicated that chest CT plays an important role in the evaluation of pulmonary injury for COVID-19 pneumonia, especially for those who are at-risk for severe pneumonia. According to the reviewer's suggestion, we discussed this issue as an important practical limitation in the revised manuscript (*Line 388-393, Page 20-21*).

Title

It is misleading to say in the Title and Abstract that the patients are from outside Wuhan, since about half of them came from or were exposed in Wuhan.

Response: Thank you for pointing out this confusing wording. We have deleted the phrase “*Outside Wuhan*” in the title, abstract and main text to avoid misunderstanding in the revised manuscript.

Abstract

Should specify the number of cases included. Should specify that the cohort was retrospective.

Response: Thank you for your kind suggestion. We have re-written the Abstract section with adding the case numbers and study design (*Line 33, Page 2*).

Introduction. It is not true that “Chest CT has been demonstrated to be an important approach for screening individuals with suspected COVID-19”. Reference 10 shows that CT is relatively insensitive for early identification of COVID-19 infection. Reference 8 addresses only CT documentation of longitudinal course in 21 patients.

Response: Thank you for pointing out this inappropriate statement and reference. A study (Ai T, et al. *Radiology*, 2020) involving 1014 cases reported that chest CT had higher sensitivity for diagnosis of COVID-19 as compared with initial RT-PCR from swab samples in Wuhan, China.² Even so, chest CT for COVID-19 screening and diagnosis is a distraction and remains controversial.^{3,4} Thus, we have re-written this sentence as “Chest CT has been proposed as an ancillary approach for screening individuals with suspected COVID-19 pneumonia during the epidemic period” and replaced the reference in the revised manuscript. (*Line 69-71, Page 3; Line 443-450, Page 23*)

Methods

The authors should indicate if any of the patients reported here were included in prior publications.

***Response:* Thank you for your reminder. None of the patients reported in this study were included in prior publications regarding their clinical characteristics. We have now added the statement to the text. (Line 84-85, Page 4)**

Were all patients directly admitted to the authors' hospital, or were some transferred from elsewhere? In particular, does the word "imported" mean that these patients were transferred from hospitals in Wuhan, or just that they were exposed in Wuhan? If they were transferred, the authors should provide the duration of admission from the date of admission in Wuhan. Please use a more specific word in place of "imported".

***Response:* The word "imported" in the original manuscript meant that they were exposed in Wuhan within 2 weeks before illness onset. During the COVID-19 outbreak, all suspected individuals were screened at fever clinic, and the infected patients were transferred to the local designated hospitals after being confirmed by CDC. Thus, all patients in our cohorts were local (some of them had traveled to Wuhan) and no case was transferred from hospitals in Wuhan. Thus, we replaced the words "imported" and "nonimported" with "exposure history in Wuhan within 2 weeks (yes/no)" to avoid misunderstanding in the revised manuscript. (Line 110-111, Page 5)**

Results

The statement "52 patients of the included patients had discharged at enrollment." Is unclear. Were these patients discharged without hospitalization?

***Response:* We are sorry for the unclear writing. All included patients with COVID-19 pneumonia were admitted to the hospital and isolated after confirmed diagnosis. In the revised manuscript, the clinical outcomes including discharge and death were recorded, and 140 and 105 patients had discharged in the derivation and validation cohorts at the time of analysis, respectively. (Line 182-183, Page 7)**

Why were patients with normal CT findings or those with severe COVID excluded?

Response: Our study aimed to explore the early clinical and imaging risk factors for progression to severe illness in patients with moderate COVID-19 pneumonia. Patients with normal CT findings did not have pneumonia but were infected with the coronavirus, who have a mild, favorable disease course and a very low risk of progression to pneumonia. Patients who had severe COVID-19 pneumonia on admission were also excluded for unavailable clinical and imaging data before admission.

Smoking history seems remarkably low compared to the prevalence of smoking in China. Is this because of incomplete data collection? How did the authors handle cases where smoking history was unknown?

Response: Thank you for your comments. It is really true as the reviewer indicated that the proportion of patients with smoking history in our cohort is lower compared to the prevalence of smoking in China.⁵ The smoking history of patients in our study was obtained from electronic medical records by trained residents. Sampling error in a relatively small population may be an explanation for the issue. Besides, several recent reports showed the proportion of COVID-19 patients with smoking history ranged from 7.0% to 14.6%.^{1, 6} The lower prevalence of smoking among the patients with COVID-19 deserves further investigation.

Likewise the age distribution appears younger than in Wuhan, particularly for the more severe cases. The authors should comment on this difference.

Response: The patients in our cohort were younger compared with those in Wuhan,⁶ which may be due to the fact that most of the patients with exposure history in Wuhan were young or middle-aged individuals working in Wuhan. They came back to Hunan province for the Chinese New Year and some of their family members who were in close contact with them were therewith infected. The patient population in Wuhan included more local elderly. The age distribution in our cohort was comparable to those in the reports by Guan et al (1099 patients from 30 provinces in China with a median age of 47 years) and Xu et al (62 patients from Zhejiang province, China with a median age of 41 years).^{1,}

⁷ We discussed this difference in the Discussion section according to the reviewer's suggestion. (Line 313-316, Page 18)

Did all patients know the date of exposure? This is surprising, as some patients may have been exposed to asymptomatic carriers.

Response: Thank you for your comments. It is true as the reviewer pointed out that the accurate source or date of exposure was not very clear for some patients in our cohort during the outbreak. However, for a non-Wuhan province in our study setting, we simply distinguished patients based on whether they had an exposure history in Wuhan within 2 weeks before illness onset. Thus, for patients who had no clear exposure history, we believed that he may have been locally exposed to other infected individuals so long as he had not been to Wuhan within 2 weeks before illness onset. (Line 110-111, Page 5)

Were other indices of disease severity measured? What about oxygen saturation, or oxygen requirement?

Response: Thank you for your kind suggestion. Oxygen saturation in the resting state is an important indicator of hypoxia or severe pneumonia. Therefore, we supplemented the percutaneous oxygen saturation on admission and compared it between stable and progressive groups. (Line 200-201, Page 9; Table 2, Page 10)

Table 1. What are the numbers provided? Some appear to be mean (or median?) with standard deviation, some appear to be mean with range.

Response: In the tables of this study, the continuous variables (such as age, lymphocyte count, CT severity score) were presented as the median and interquartile range (IQR) because most of these variables were non-normal distribution, and categorical variables (such as gender, comorbidities) were presented as frequency and percentage. We have added these instructions to the tables in the revised manuscript.

Figure 3. Unclear what the shaded markings, and the white rectangles refer to. Please explain in figure legend. Please also include a marker of the mean and standard deviation.

Response: We are sorry for this. We have supplemented the explanation for the shaded markings and the white rectangles in figure legends (*Supplementary Figure 1, Page 28*). Since the CT severity scores were non-normal distribution, we marked the median and interquartile range (IQR) in the revised figure.

Figure 4. I don't think that this conveys much useful information, since the correlations between imaging and non-imaging features

Response: COVID-19, caused by SARS-CoV-2, involves multiple organs or systems (including gastrointestinal tract⁸, liver⁹, cardiovascular system¹⁰, and nervous system¹¹) with the lungs as the major attack target. Lung involvement reflects the most serious degree of damage caused by the coronavirus on various organ systems, while the inflammation indexes mentioned in this study represent the levels of immune-inflammatory response secondary to these injuries. Thus, the correlations between imaging and non-imaging features in our study provided the association between organ damage (mainly lung) and inflammatory response, and we further found that the pulmonary injury on admission had the potential to can early predict lymphopenia, which is an important indicator for deterioration to severe pneumonia. (*Line 258-267 and Figure 4, Page 15-16*)

References

1. Guan WJ, *et al.* Clinical Characteristics of Coronavirus Disease 2019 in China. *N Engl J Med*, (2020).
2. Ai T, *et al.* Correlation of Chest CT and RT-PCR Testing in Coronavirus Disease 2019 (COVID-19) in China: A Report of 1014 Cases. *Radiology* **0**, 200642 (2020).
3. Hope MD, Raptis CA, Shah A, Hammer MM, Henry TS. A role for CT in COVID-19? What data really tell us so far. *The Lancet*, (2020).
4. Wang YXJ, Liu W-H, Yang M, Chen W. The role of CT for Covid-19 patient's management remains poorly defined. *Annals of Translational Medicine* **8**, 145 (2020).
5. Wang M, *et al.* Trends in smoking prevalence and implication for chronic diseases in China: serial national cross-sectional surveys from 2003 to 2013. *The Lancet Respiratory medicine* **7**, 35-45 (2019).
6. Chen T, *et al.* Clinical characteristics of 113 deceased patients with coronavirus disease 2019: retrospective study. *BMJ* **368**, m1091 (2020).

7. Xu X-W, *et al.* Clinical findings in a group of patients infected with the 2019 novel coronavirus (SARS-Cov-2) outside of Wuhan, China: retrospective case series. *BMJ* **368**, m606 (2020).
8. Gu J, Han B, Wang J. COVID-19: Gastrointestinal Manifestations and Potential Fecal-Oral Transmission. *Gastroenterology*, (2020).
9. Fan Z, *et al.* Clinical Features of COVID-19 Related Liver Damage. *medRxiv*, 2020.2002.2026.20026971 (2020).
10. Zheng YY, Ma YT, Zhang JY, Xie X. COVID-19 and the cardiovascular system. *Nat Rev Cardiol*, (2020).
11. Mao L, *et al.* Neurological Manifestations of Hospitalized Patients with COVID-19 in Wuhan, China: a retrospective case series study. *medRxiv*, 2020.2002.2022.20026500 (2020).

REVIEWER COMMENTS

Reviewer #1 (Remarks to the Author):

I think that you have done a very good job answering most of the queries

Reviewer #2 (Remarks to the Author):

The authors have added substantially more data to the manuscript including a validation cohort. Overall this has strengthened the paper.

The major weaknesses remain the clinical relevance of a prediction model heavily dependent on chest CT in healthcare systems which are actively trying to avoid this intervention. The second issue is the clinical relevance of predicting "severe" pneumonia by their criteria.

The additional information also creates some additional questions. How do the authors explain the paradoxical association of bacterial superinfection with less frequent consolidation on chest CT scan.

The statements that "coronavirus consumes many immune cells" and "reduced but hyperactivated peripheral T cells" (line 324-7) account for the severe lung injury is conjecture and not backed up by data. The NLR itself is problematic since elevation can either be increased neutrophils or decreased lymphocytes or relative changes in both. COVID-19 patients could have all three.

Reviewer #3 (Remarks to the Author):

The authors have done a good job of responding to the initial critique. In particular, the inclusion of a new validation cohort, and development of a predictive nomogram with ROC analysis are very helpful additional features. Since the pace of publications on COVID-19 is remarkably fast, there are a couple of new points to address, in addition to residual issues from the prior critique.

Specific comments

I believe that the nomogram is a very helpful feature of this paper. Is it possible to convert this to an online risk calculator?

Response to reviewer #1, point #4 is not sufficient. There is emerging evidence that the severity of lung disease measured by chest radiograph is a strong predictor of outcome. In a recently published study, the chest radiograph emerged as an important risk factor for occurrence of critical illness (JAMA Intern Med. 2020 May 12. doi: 10.1001/jamainternmed.2020.2033.) In this study, the CT was not included in the final model, presumably because the predictive value of the CXR score outweighed the strength of the CT score. The authors should acknowledge and address this issue, and should compare their analysis with the analysis in this paper.

Response to reviewer #2, minor point #1. The statement that "Early identification of patients at risk of severe pneumonia is of clinical importance to reduce the case fatality rate" is still present in the

abstract, and not supported by any data. A similar statement is present in the conclusion of the abstract, and in the conclusion of the discussion section. Please delete.

Also, the higher severity score in those admitted more than 4 days after symptom onset is likely due to lead time bias, since it is known that the severity of disease increases until about 10 days after symptom onset (<https://doi.org/10.1148/radiol.2020200370>). Thus, this finding does not mean that delayed admission results in greater disease severity. The authors should critically evaluate and discuss the potential role of lead time bias in contributing to their findings. Should the risk score be adjusted for days since symptom onset?

Line 59- should read “prevalence of severe COVID-19 ranged from 15.7% to 26.1% among patients admitted to hospital”.

Line 72- should indicate this was retrospective evaluation

Line 172- was there a second exclusion criterion? The numbers go from 1 to 3.

The authors should compare their results with the just-published analysis Development and Validation of a Clinical Risk Score to Predict the Occurrence of Critical Illness in Hospitalized Patients With COVID-19 (JAMA Intern Med. Published online May 12, 2020. doi:10.1001/jamainternmed.2020.2033). In this article, a CT severity score was not included in the final risk model, although chest radiographic abnormality remained in the model. This might be because the authors included a wider range of features (n=72) in their LASSO analysis. The authors should discuss why this difference might be.

Point-by-point answers to the reviewers' comments.

Their comments are reproduced and our responses are given afterward in bold.

All changes in the main text are in red color.

Reviewer #1 (Remarks to the Author):

I think that you have done a very good job answering most of the queries

Response: We thank the reviewer for the approval of the manuscript.

Reviewer #2 (Remarks to the Author):

The authors have added substantially more data to the manuscript including a validation cohort. Overall this has strengthened the paper.

The major weaknesses remain the clinical relevance of a prediction model heavily dependent on chest CT in healthcare systems which are actively trying to avoid this intervention. The second issue is the clinical relevance of predicting "severe" pneumonia by their criteria.

Response: Thank you for your recognition of our revision, and we appreciate your insightful comments. The prediction model proposed in our study included CT severity score, which is used to quantitatively estimate the degree of lung injury and is associated with both the number of involved lobes and the extent of lesions. Lung injury is the cornerstone in the pathophysiology of COVID-19, which represents the most serious degree of damage caused by the coronavirus on various organ systems. In addition, systemic inflammation and multiple organ dysfunction during the exacerbation to severe or even critical illnesses are also secondary to lung injury. Chest CT is superior over X-ray in showing the extent and characteristics of pulmonary lesions. Thus, we recommend chest CT examination after admission and X-ray as a monitoring tool during treatment. Given the fact that CT is mainly used for hospitalized, symptomatic patients with specific clinical indications in real-world practice, we also pointed out this application limitation in the discussion. (Line 415-420, Page 21)

Our study aimed to early predict the in-hospital progression risk within 14 days in patients with moderate COVID-19 pneumonia on admission. Severe illness is

characterized by hypoxia according to our clinical severity classification of COVID-19 pneumonia, which may induce cytokine storm and acute cardiovascular events, and need oxygen therapy. Thus, early prediction of the risk of severe pneumonia in our study has the potential to facilitate more individually aligned treatment plans, optimize utilization of medical resource, and prevent further deterioration.

The additional information also creates some additional questions. How do the authors explain the paradoxical association of bacterial superinfection with less frequent consolidation on chest CT scan.

***Response:* As indicated by the reviewer, our results showed that patients who developed bacterial co-infection during hospitalization were less likely to show consolidation on CT at the time of admission ($P = 0.012$) compared to those without bacterial co-infection. Most of the patients who developed nosocomial bacterial infection are generally the elderly, long-term bedridden, and immunocompromised. Likewise, in our study, patients who developed in-hospital pulmonary bacterial co-infection were significantly older and more likely to have underlying hypertension. In this case, existing co-morbidities, plus impaired lung function and dysregulated immune responses that occur with even healthy aging may weaken the antiviral immune response, resulting in low-level pulmonary mesenchymal inflammatory cells infiltration and exudates in the alveolar cavity (reduced necrosis and protein components) at the early stage of COVID-19 pneumonia. This may explain the potential association of bacterial co-infection with less frequent consolidation on baseline chest CT. Even so, we also acknowledged that our findings were limited by the relatively small sample size, which should be interpreted with caution by clinicians and further confirmed by larger samples. (Line 397-399, Line 401-404, Page 20-21)**

The statements that "coronavirus consumes many immune cells" and "reduced but hyperactivated peripheral T cells" (line 324-7) account for the severe lung injury is conjecture and not backed up by data. The NLR itself is problematic since elevation can either be increased neutrophils or decreased lymphocytes or relative changes in both. COVID-19 patients could have all three.

Response: Thank you for your comments. The statements were our hypothesis on the basis of some previous reviews, while were also supported by some recently published articles. In order to state our viewpoint more modestly, we would rewrite this passage in the revised manuscript as follow: “Some studies suggested that the decrease of peripheral T lymphocyte count attributes to the inflammatory cytokine milieu and T cell recruitment to sites of infection, and reduced but hyperactivated or exhausted peripheral T cells were more frequently found in severe cases. Lymphopenia has been confirmed as a potential factor associated with disease severity and mortality in COVID-19. Thus, damage to lymphocytes and consequently immunologic abnormality might be an important factor leading to exacerbations of patients. The uncontrolled inflammatory response could also stimulate the production of neutrophils apart from speeding up the apoptosis of lymphocytes. NLR, a simple biomarker to assess the systemic inflammatory status, is widely used for the prediction of prognosis of patients with pneumonia. Increased NLR, resulting from decreased lymphocyte count and/or elevated neutrophil count, represents damaged lymphocyte function and/or increased inflammatory level and risk of bacterial infection.” (*Line 334-347, Page 18-19*)

Reviewer #3 (Remarks to the Author):

The authors have done a good job of responding to the initial critique. In particular, the inclusion of a new validation cohort, and development of a predictive nomogram with ROC analysis are very helpful additional features. Since the pace of publications on COVID-19 is remarkably fast, there are a couple of new points to address, in addition to residual issues from the prior critique.

Specific comments

I believe that the nomogram is a very helpful feature of this paper. Is it possible to convert this to an online risk calculator?

Response: Thank you for your recognition of our revision and kind suggestions for further improve our manuscript. We have converted the nomogram into an online risk calculator (<https://xy3yyfskzfc.shinyapps.io/DynNomapp2/>). This has been indicated in the revised manuscript. (*Line 247-248, Page 13*)

Response to reviewer #1, point #4 is not sufficient. There is emerging evidence that the severity of lung disease measured by chest radiograph is a strong predictor of outcome. In a recently published study, the chest radiograph emerged as an important risk factor for occurrence of critical illness (JAMA Intern Med. 2020 May 12. doi: 10.1001/jamainternmed.2020.2033.) In this study, the CT was not included in the final model, presumably because the predictive value of the CXR score outweighed the strength of the CT score. The authors should acknowledge and address this issue, and should compare their analysis with the analysis in this paper.

Response: Thank you for your suggestions. Our results in response to reviewer #1, point #4 showed that the sensitivity of chest radiograph was insufficient to identify the presence of pneumonia and there was a high risk of missed diagnosis. However, chest radiograph can be used to monitor the progression of pulmonary involvement after confirmed diagnosis, and has emerged as an important risk factor for the occurrence of critical illness by Liang et al.¹ In their study, the CT was not included in the final model. There were several characteristics in their study, including 1) all hospitalized patients with confirmed COVID-19 were included; 2) critical illness or death was defined as the composite endpoint; 3) the variables regarding chest radiograph or CT abnormalities were dichotomous (yes vs. no). These settings made that the predictive value of chest radiograph outweighed the strength of CT. However, we adopted a quantitative CT severity score (range: 0-25) to accurately assess the degree of lung injury and aimed to early predict the in-hospital progression risk (or development of severe illness) within 14 days in patients with moderate COVID-19 pneumonia on admission. In addition, the prediction model established in our study was simpler with only three easily accessible variables compared with theirs. We have now added these discussions in the revised manuscript. (Line 378-385, Page 20)

Response to reviewer #2, minor point #1. The statement that “Early identification of patients at risk of severe pneumonia is of clinical importance to reduce the case fatality rate” is still present in the abstract, and not supported by any data. A similar statement is present in the conclusion of the abstract, and in the conclusion of the discussion section. Please delete.

Response: We thank the reviewer for pointing out these inappropriate statements. We would revise the statement in the background of the abstract as “Early

identification of hospitalized patients at risk of progression is of clinical importance and may facilitate more individually aligned treatment plans and optimized utilization of medical resource”. (Line, Page 2) Likewise, we also modified the statements in the conclusion of the abstract and discussion sections according to the reviewer’s suggestion. (Line 31-33, Line 43-44, Page 2; Line 320-322, Page 18)

Also, the higher severity score in those admitted more than 4 days after symptom onset is likely due to lead time bias, since it is known that the severity of disease increases until about 10 days after symptom onset (<https://doi.org/10.1148/radiol.2020200370>). Thus, this finding does not mean that delayed admission results in greater disease severity. The authors should critically evaluate and discuss the potential role of lead time bias in contributing to their findings. Should the risk score be adjusted for days since symptom onset?

Response: Thank you for the constructive comment, and we discussed this issue with a senior Professor of Statistics (Hongzhuan Tan, Department of Epidemiology & Health Statistics, Xiangya School of Public Health, Central South University). Our results showed that for patients who were admitted more than 4 days from symptom onset, more lobes or segments involved, and higher CT severity scores were found. Lead time bias refers to the phenomenon where early diagnosis of a disease or admission falsely makes it look like people are surviving longer. However, our study focused on the probability of specific outcome happening rather than the survival time. Namely, when patient A who was early admitted (such as 1 day after illness onset) died 10 days after admission and patient B who was late admitted (such as 8 day after illness onset) died 3 days after admission, the conclusion that delayed admission was associated with shorter survival time is due to lead time bias. Thus, our conclusion should be reasonable, since lung injury in patients with COVID-19 pneumonia after symptoms onset gradually extended and aggravated, and our study revealed that CT severity score was an independent risk factor for short-term progression. Pan et al also reported that in patients recovering from non-severe COVID-19, the severity of lung abnormalities gradually increased until approximately 10 days after initial onset of symptoms.² According to our findings, early admission and timely intervention reversed the progression of

lung injury, thereby reducing the risk of adverse outcomes. In addition, the CT severity score was closely associated with days since symptom onset, thus the risk score was not adjusted for days since symptom onset.

Line 59- should read “prevalence of severe COVID-19 ranged from 15.7% to 26.1% among patients admitted to hospital”.

Response: Thank you for your suggestion. We have modified this sentence as recommended. (Line 62, Page 3)

Line 72- should indicate this was retrospective evaluation

Response: Thank you for your reminder. We have now added the statement to the text. (Line 75, Page 3)

Line 172- was there a second exclusion criterion? The numbers go from 1 to 3.

Response: Thank you for pointing out this error. Th exclusion criteria only included two items as the revised manuscript shown. (Line 180, Page 7)

The authors should compare their results with the just-published analysis Development and Validation of a Clinical Risk Score to Predict the Occurrence of Critical Illness in Hospitalized Patients With COVID-19 (JAMA Intern Med. Published online May 12, 2020. doi:10.1001/jamainternmed.2020.2033). In this article, a CT severity score was not included in the final risk model, although chest radiographic abnormality remained in the model. This might be because the authors included a wider range of features (n=72) in their LASSO analysis. The authors should discuss why this difference might be.

Response: As the reviewer indicated, Liang et al developed and validated a clinical risk score at hospital admission for predicting the occurrence of critical illness, which incorporating 10 variables selected from 72 potential predictors by using LASSO and logistic regression analyses.¹ However, their risk score included chest X-ray abnormality instead of the severity of abnormality on CT. There were several differences between our study and theirs, including 1) we enrolled only patients with moderate COVID-19 pneumonia at the time of admission, while they included all hospitalized patients with confirmed

COVID-19 for model development; 2) the primary endpoint in our study was the development of severe illness within 14 days after admission while they defined critical illness or death as the composite endpoint, which indicated that we focused on the earlier progression; 3) they included a wider range of features in the LASSO analysis but the variables regarding chest radiograph or CT abnormalities were dichotomous, while we adopted a quantitative CT severity score to accurately assess the degree of lung injury. Compared with their study, the prediction model in our study was simpler with only three easily accessible variables, which achieved comparable performance. We have now added these discussions in the revised manuscript. (Line 378-385, Page 20)

1. Liang W, *et al.* Development and Validation of a Clinical Risk Score to Predict the Occurrence of Critical Illness in Hospitalized Patients With COVID-19. *JAMA Intern Med*, (2020).
2. Pan F, *et al.* Time Course of Lung Changes at Chest CT during Recovery from Coronavirus Disease 2019 (COVID-19). *Radiology* **295**, 715-721 (2020).

REVIEWERS' COMMENTS:

Reviewer #2 (Remarks to the Author):

The authors have adequately addressed my previous concerns. I have no further comments/recommendations.

Reviewer #3 (Remarks to the Author):

The authors have done an excellent job of responding to most of the remaining comments. However, the statements that "early admission and CT examination should be recommended." (line 44/45), "early admission to avoid severe pulmonary damage and CT examination may be important for the individual management of COVID-19 pneumonia patients" (Line 406-408) and "Early admission is required for avoiding severe pulmonary damage for COVID-19 pneumonia patients" (424-427) are still not supported by the data. The disease progresses up to 10-14 days, and the authors present no evidence that early admission improves prognosis or prevents this progression.

I have checked the online nomogram, and its performance is variable. It worked OK initially, but subsequent entry of different data produced an error.

Point-by-point answers to the reviewers' comments.

Their comments are reproduced and our responses are given afterward in bold.

All changes in the main text are highlighted by using the track changes.

REVIEWERS' COMMENTS:

Reviewer #2 (Remarks to the Author):

The authors have adequately addressed my previous concerns. I have no further comments/recommendations.

Response: We thank the reviewer for the approval of the manuscript.

Reviewer #3 (Remarks to the Author):

The authors have done an excellent job of responding to most of the remaining comments.

However, the statements that "early admission and CT examination should be recommended." (line 44/45), "early admission to avoid severe pulmonary damage and CT examination may be important for the individual management of COVID-19 pneumonia patients" (Line 406-408) and "Early admission is required for avoiding severe pulmonary damage for COVID-19 pneumonia patients" (424-427) are still not supported by the data. The disease progresses up to 10-14 days, and the authors present no evidence that early admission improves prognosis or prevents this progression.

Response: Thank you for your recognition of our revision, and we appreciate your insightful comments. We agreed with the perspective of the reviewer and found that these statements "early admission should be recommended to avoid severe pulmonary damage" were indeed not rigorous based on our data. As the reviewer indicated, the association between late admission and more severe pulmonary involvement may be attributed to the natural history of disease progression during the first 10-14 days, while we hold the opinion that the prediction of progression risk within 14 days of COVID-19 pneumonia patients using chest CT and clinical characteristics may also provide an important basis for guiding clinicians to determine the rational timing of admission. Thus, we rewrote these sentences in the revised manuscript as follows:

1) CT examination may help the prediction of progression risk and guide the

timing of admission. (*Line 41-43, Page 2*)

2) CT examination may be important in guiding the rational timing of admission for the individual management of COVID-19 pneumonia patients. (*Line 277-279, Page 11*)

3) Chest CT has the potential to early predict the risk of progression and reflect disease severity as well, which may also help guide the timing of admission for COVID-19 pneumonia patients. (*Line 295-298, Page 11*)

I have checked the online nomogram, and its performance is variable. It worked OK initially, but subsequent entry of different data produced an error.

***Response:* We are sorry for the variable performance of our online nomogram. To ensure the stability of the online tool, we have upgraded our shinyapps account. We then checked the online nomogram again and confirmed it worked well.**